# Which Diet Has the Least Environmental Impact on Our Planet? A Systematic Review of Vegan, Vegetarian and Omnivorous Diets

**Bingli Clark Chai, Johannes Reidar van der Voort, Kristina Grofelnik, Helga Gudny Eliasdottir, Ines Klöss and Federico J. A. Perez-Cueto \***

Design & Consumer Behaviour Section, Department of Food Science, Faculty of Science, University of Copenhagen, Rolighedsvej 26, 1958 Frederiksberg C, Denmark
\* Correspondence: apce@food.ku.dk

**Abstract:** The food that we consume has a large impact on our environment. The impact varies significantly between different diets. The aim of this systematic review is to address the question: Which diet has the least environmental impact on our planet? A comparison of a vegan, vegetarian and omnivorous diets. This systematic review is based on 16 studies and 18 reviews. The included studies were selected by focusing directly on environmental impacts of human diets. Four electronic bibliographic databases, PubMed, Medline, Scopus and Web of Science were used to conduct a systematic literature search based on fixed inclusion and exclusion criteria. The durations of the studies ranged from 7 days to 27 years. Most were carried out in the US or Europe. Results from our review suggest that the vegan diet is the optimal diet for the environment because, out of all the compared diets, its production results in the lowest level of GHG emissions. Additionally, the reviewed studies indicate the possibility of achieving the same environmental impact as that of the vegan diet, without excluding the meat and dairy food groups, but rather, by reducing them substantially.

**Keywords:** omnivorous; vegan; vegetarian; sustainable; diet; GHG; LCA; carbon; environment; footprint; systematic review

## 1. Introduction

According to the United Nations, the world's population will grow to 9.8 billion people by 2050 [1]. This corresponds to an increase of almost 30% from the current population of 7 billion [2]. Demographic changes and population growth imply an increasing demand for animal products, particularly meat, dairy products and crops, which suppliers need to fulfil [2]. It is expected that by 2050, milk and meat production will increase 58 and 73 percent, respectively [3]. Concerns about animal welfare have been an issue for centuries, but climate change and greenhouse gas emissions (GHGEs) have recently started to become a point of interest [4]. Modern food systems, especially the agriculture sector, have a highly unsustainable impact on the environment. Using natural resources (land, water and fossil energy) to raise livestock and produce crops increases environmental degradation day by day. Agriculture alone is responsible for fully 10–12% of global GHGEs. It is estimated that GHGEs will rise by up to 150% of current emission levels by 2030 [5]. For this reason, finding ways to mitigate the negative impact of climate change and the environmental footprint of the current food system's environmental footprint is becoming more and more urgent. A sustainable diet is one with production that has little environmental impact, is protective and respectful of biodiversity and of ecosystems, and is nutritionally adequate, safe, healthy, culturally acceptable and economically affordable [6,7].

Animal husbandry requires a large number of inputs. At the same time, it generates undesirable by-products that affect the environment. While there is no universally accepted system for measuring these effects quantitatively, a widely used method is the Life Cycle Impact Assessment technique (LCAs) [8]. LCAs can estimate the environmental impacts of production, transport, processing, storage, waste disposal and other life stages of food production [9]. This paper will use an LCA approach to frame and systematically review the literature as related to the effects of food production on three main indicators: GHGEs, land use and the water footprint. We will evaluate variations in these effects that are associated with production for three diets, which differ mainly by their consumption of animal based products: vegan diets do not include any products from animal origin; lacto-ovo-vegetarian diets (LOV) include milk, dairy products and eggs, but no animal meat; and omnivorous diets (OMN) includes all animal-based food products, including meat, dairy and eggs.

## 2. Methods

### 2.1. Literature Search Strategy

In order to find articles relevant to the research question, the electronic bibliographic databases PubMed, Medline, Scopus and Web of Science were searched. The literature search took place on 9 April 2019. The specific search strategy and careful selection of the terms (environment* AND diet* AND (footprint OR sustain* OR impact) AND ("greenhouse gas*" OR vegetarian* OR vegan OR omnivorous) AND diet* AND (footprint OR sustain* OR impact) AND ("greenhouse gas*" OR vegetarian* OR vegan OR omnivorous) AND (water) AND (land)) AND (vegetarian OR vegan) AND (Life cycle analysis)) were entered and 1246 hits were obtained. The language was restricted to English. In order to include relevant articles from older years, no time restriction was set. Moreover, no citation-based search of prior references to relevant articles was used. That is, the literature search strategy was limited to a direct database search. In addition to work identified through this process, two additional articles provided by a senior researcher have been included in the review. The keywords used in the literature search are based on a joint agreement between the junior and senior authors.

### 2.2. Inclusion and Exclusion Criteria

The included studies were selected by focusing directly on keywords related to the environmental impacts of diets, rather than on special nutritional differences and health effects for the human body. Some of the reviewed articles combine health aspects and the environmental footprint. Only articles about human dietary patterns were included. Scientific papers on specific dietary patterns such as: the New Nordic Diet, the Mediterranean Diet or the Atlantic Diet have been excluded, because it is difficult to classify these diets as fully omnivorous, vegetarian or vegan. Articles that report research performed on a very specific target group (e.g., Carbon footprint and land use of conventional and organic diets in Germany, Treu Hanna et al., 2017.) were also excluded. Moreover, articles that focused on technological improvements in agriculture, economic analyses on food systems and sociological aspects behind food choice/consumer behaviour were excluded.

### 2.3. Screening

After the initial list was generated, duplicates were removed using Excel. Next, the titles of all studies were screened by five reviewers, to avoid reflexivity bias; titles that did not fulfil the inclusion criteria were removed. Further screening was conducted by reading each full-text article, focusing on our criteria.

### 2.4. Data Extraction

In order to extract relevant data from the selected 34 articles, a predetermined grid was created and served as a tool to organize an overview of all relevant information. Two tables were created, one for all studies and the other for all reviews. Each table was split up into the following categories:

author, publication year, country, diet comparison, quality assessment, main outcome, and magnitude. The table of studies also included space for a description of the study design and for duration. These data are presented here as Tables 1 and 2.

### 2.5. Quality Assessment

Methodological and reporting quality assessment was performed in order to evaluate the robustness of the conclusions of the review. The quality assessment was done by checking each of the used research articles and reviews according to a predetermined set of criteria (Tables 3 and 4) and following Research Connections Quantitative Quality Assessment Tool (https://www.researchconnections.org/content/childcare/understand/research-quality.html, retrieved at 16 January 2019) and National Heart, Blood and Lung Institute Study Quality Assessment Tool (https://www.nhlbi.nih.gov/health-topics/study-quality-assessment-tools, retrieved 16 January 2019). Additionally, all authors had to achieve consensus on the final mark. Articles and reviews were managed by using the software Mendeley. The papers were categorized in low quality, medium quality and high quality. This leads to 9 low quality papers, 9 medium quality papers and 16 high quality papers, with a total of 34 papers.

**Table 1.** Characteristics of the included studies.

| Authors/Year | Country | Study Design | Description of Intervention | Duration | Diet Comparison | Quality Assessment | Main Outcome |
|---|---|---|---|---|---|---|---|
| Harwatt, H./2017 | USA | LCA | Simple analysis, which replacement of food could be a contributor to achieve GHGEs reduction. | N/A | comparing omnivorous diet and plant-based diet | Medium | Replacing beans for beef in general diets could achieve a reduction of cropland by 42% and 46%–74% reduction of GHGEs need by the 2020 target in the US. |
| Hyland, J./2016 | Ireland | Descriptive Analysis | Using data from the National Adult Nutrition Survey in Ireland to analyze GHGEs for the total population and various categories | 2008–2010 | comparing omnivorous, vegetarian and plant-based diet | High | Highest contributor to GHGEs was meat with 1646 g COs-eq. The second largest daily emissions were dairy and starchy products with 732 g COs-eq and lowest were vegetables, fruits and legumes with 647 COs-eq. |
| Pradhan, P./2016 | Global | Global data analysis | Global analysis of 16 different global dietary patterns | 1961–2007 | 16 dietary patterns based on energy content of these diets | Medium quality | Highlighting the changes in food consumption over the past 50 years and their regionality. |
| Rosi, A./2017 | Italy | LCA | Real-life context. Controlled intervention among 3 designed diets groups. | 7 consecutive days | omnivorous, vegan, lacto-ovo-vegetarian | High quality | It shows a pattern that vegan diet is better than omnivorous diet in terms of environmental footprint. |
| Ulaszewska, M.M./2017. | Italy | LCA | Comparison of Mediterranean and New Nordic diet in terms of GHGEs | N/A | Mediterranean and New Nordic diet | Low quality | GHGEs for high-protein and vegetable/fruit group in recommendations is comparable and similar |
| Tyszler, M./2016 | Netherlands | LCA and questionnaire | Effect of different variations of the current diet on the environment with comparison with vegetarian and vegan diet | N/A | Vegan, vegetarian, current and closest healthy diet | Medium quality | There is a diet, not much different from the current Dutch diet, that has the same effect on the environment as the vegan diet |
| Arrieta, E.M./2018 | Argentina | Scenario analysis | Estimating the GHGEs of different diets in Argentina through a scenario analysis | N/A | National diet, diet with no ruminant meats, LOV and vegan diet | Low quality | Least GHGEs from vegan diet, highest GHGEs from national Argentinian diet. |
| Meier, T./2013 | Germany | LCA | Comparison of environmental impact of 4 dietary scenarios in the period between 1985–1989 and 2006. | N/A | D-A-CH UGB LOV vegan | High quality | All the indicators of environmental impact are lower in 2006, compared to 1985–1989. because of the change in diet |

**Table 1.** *Cont.*

| Authors/Year | Country | Study Design | Description of Intervention | Duration | Diet Comparison | Quality Assessment | Main Outcome |
|---|---|---|---|---|---|---|---|
| Blackstone, N.T./2018 | USA | LCA | Comparing different diets and estimating their environmental impact | N/A | Healthy US diet, healthy Mediterranean diet and healthy vegetarian diet | Low quality | A healthy vegetarian diet has 84–42% lower climate impacts than the healthy US diet and Mediterranean diet with the exception of water use which was the same. |
| Seconda, L./2018 | France | Questionnaire | Evaluating different dietary patterns to assess their environmental impact and characterizing consumer dietary patterns | N/A | Consumer dietary patterns | High quality | Dietary patterns among the consumers were not seen as sustainable and more sustainable diets contained less meat and less processed food. |
| Corrado et al./2019 | Italy | LCA | Evaluating different LCA associated with three dietary patterns | N/A | Vegan, vegetarian and omnivorous | Medium quality | A reduction in the GHG emission would be attained by changing the dietary patterns to vegan and vegetarian under certain limits |
| Esteve-Llorens et al./2019 | Spain | LCA | Evaluating the carbon footprint through life cycle by analyzing the Atlantic diet | N/A | Omnivorous and Atlantic diet | Medium quality | Atlantic dietary is beneficial from both health and environmental perspective |
| Green et al./2018 | UK | GHGEs and water footprint, LCA | Evaluating the environmental footprint in agriculture (India) | N/A | Omnivorous, vegan, fruitirism, vegetarian | Medium quality | Environmental impact of certain diets in India are relatively low compared with high income countries |
| Van Dooren, C./2014 | Netherlands/global | LCA | Analyzing six different dietary patterns to assess their nutritional value and environmental impact | N/A | Average Dutch diet, recommended Dutch diet, semi-vegetarian, vegetarian, vegan and Mediterranean | Medium quality | High health scores of diets are linked to high sustainability scores. The vegan diet has the highest sustainability score while the Mediterranean diet has the highest health score. |
| Weber, C./2008 | USA | Method: IO-LCA | LCA of GHGEs associated with distribution N/A | N/A | No comparison | High quality | Transport contributes to only 11% of GHGEs. Delivery from producer to retail contributes only with 4% |
| Scarborough, P./2014. | UK | FFQ | Are there differences in different diets contribution to GHGEs | N/A | Omnivorous, vegan, vegetarian, fish-eaters | High quality | GHGEs are twice as high in meat eaters as those in vegans |

**Table 2.** Characteristics of the included reviews.

| Author/Year | Country | Description of Review | Diet Comparison | Quality Assessment | Main Outcomes |
|---|---|---|---|---|---|
| González-García, S./2018 | Spain | Systematic analysis of 21 Peer-Review Studies | Examines 66 dietary scenarios and their carbon footprint, including vegetarian, vegan and omnivorous diet patterns | High quality | Dietary choices have higher carbon footprints if they are meat-rich; reducing animal products is advantageous for the environment |
| Wanapat, M./2015 | Global | Different feed additive practices for ruminants | No comparison | High quality | Decrease of methane production from ruminants will contribute to reduction of global methane production |
| Garnett, T./2013 | Global | How to make food production more environmentally sustainable and resilient while feeding more people more effectively | No comparison | Low quality | The priority for the future should be a nutrition-driven food system that sits within environmental limits. |
| Pimentel, D./2003 | USA | Comparison between meat-based diet and lacto-ovo-vegetarian diet in terms of environmental footprint. | Meat-based diet and lacto-ovo-vegetarian diet | Low quality | In the long term, both diets are not sustainable. However, the meat-based diet uses more resources than lacto-ovo-vegetarian diet. Therefore, between them, lacto-ovo-vegetarian diet is more sustainable. |
| Reijnders, L./2003 | USA | Quantitative evaluation of different types of protein sources. Comparison of different types of protein and their emissions. | Vegetarian and non-vegetarian. | High quality | Encouraging individuals to eat more efficiently on the food chain where they consume less meat and more plant-based, will reduce the environmental cost of food production. |
| Ridoutt, B./2017 | Australia | Environmental impact of different diets | Vegetarian, vegan and non-vegetarian. | High quality | In general, average diets have a higher emission on environmental aspects than recommended diets. |
| Sabaté, J./2014 | Global | Comparison of plant-based and animal-based diets in terms of environmental impact | Plant-based and animal-based diet | Medium quality | Implementing plant-based diet is the best option for sustainable future |
| Friel, S./2009 | UK | Agricultural strategies to reduce emissions by 80% till 2050 | No comparison | Medium quality | Formulation of policies that consider equitable distribution and reduction of livestock production is needed |
| Aleksandrowicz, L./2016. | Global | Different diet types and their effect on GHGEs, land and water use | Comparison of plan-based diets, omnivorous diets and their variations | High quality | Shift to more sustainable diet variations can show reduction of 50% water use and 70% land use and GHGEs |

**Table 2.** *Cont.*

| Author/Year | Country | Description of Review | Diet Comparison | Quality Assessment | Main Outcomes |
|---|---|---|---|---|---|
| Jones, A.D./2016. | Global | What are the components of sustainability and how are they measured | Comparison of diets considered sustainable | High quality | 3 different approaches for defining sustainable diets |
| Heller, M.C./2013 | USA | Need to combine nutrition assessment and life cycle assessment | No comparison of diets | High quality | Nutritional quality indices |
| Hess, T./2015 | UK | Different GHGEs and water use for different starchy carbohydrate sources | No comparison of diets | Low quality | Rice has the biggest impact on the environment, followed by pasta and potatoes |
| Niles, M.T./2018 | Global | Review on the possibilities for mitigating climate change in the food chain | No comparison of diets | Low quality | Non-ruminant meat consumption will lead to lower GHGEs. |
| Van Kernebeek, H.R.J./2014 | Global | Review of 12 LCA studies to study the environmental impact of human diet | Human diets with varying degrees of animal-source food products | High quality | Higher intake of animal products led to a higher intake of protein and higher intake of animal-based protein has a bigger environmental impact |
| John Reynolds, C./2014 | Global | Review of dietary advice from the World Health Organization and its environmental impact. | Diets with, reduced fat consumption, reduced animal-based food consumption and increased fruit and vegetable consumption | High quality | Reducing animal-based food consumption and increasing fruit and vegetable consumption decreases the environmental impact of consumption. Decreasing the amount of dietary fat has little to no effect on the environmental impact. |
| Cleveland, D./2017 | USA | What is the contribution of plant-based diets to climate change. | Vegan, lacto-ovo-vegetarian and omnivorous diet | Low quality | Most plant-based diets have a much lower GHGE than omnivorous diets, they are important in preventing climate change. The food industry needs to change and motivating diet change is a huge challenge. |
| Tilman, D./2014. | Global | Quantification of global diets in connection with environmental impact | Vegetarian, pescatarian, Mediterranean and omnivorous diet | Low quality | Offer of different scenarios that could help lower environmental impact of diets. |
| Gerber, P.J./2013. | Global | What is contribution of livestock production to global emission of GHGs | No diets compared | High quality | By improving technology, emissions coming from livestock production could be reduced. There is a need to make strategies for developing countries. |

**Table 3.** Quality assessment criteria of included studies.

| Low Quality | Medium Quality | High Quality |
|---|---|---|
| POPULATION AND SAMPLE:<br><br>• No description of the sample selection procedure<br>• No information on response rate or participation rate<br>• Sample size smaller than similar study or sample size not given | POPULATION AND SAMPLE:<br><br>• Non-random selection<br>• Sample size the same as similar studies<br>• Moderate to low response rate (less than 65%)<br>• Population represents a limited, atypical or selective subgroup of the population of interest | POPULATION AND SAMPLE:<br><br>• Randomized control studies<br>• Sample size larger than similar studies<br>• Participation response rate high (65–100%)<br>• Eligible population include entire population of interest or a substantial portion of it<br>• Valid (internal/external/construct) |
| MEASUREMENT:<br><br>• Main variables or concepts are not defined<br>• No discussion of variable operationalization | MEASUREMENT:<br><br>• Poor definition of main variables/concepts or it cannot be matched<br>• Measurement of key concepts with variables that do not make sense | MEASUREMENT:<br><br>• Description is accurate and can be matched<br>• Measurement of key concepts with variables that make sense |
| ANALYSIS:<br><br>• No presentation of means and standards deviations<br>• No discussion on missing data<br>• No explanation of statistical techniques | ANALYSIS:<br><br>• Standard deviations are not presented but means are<br>• Cases with missing data are removed from the analysis<br>• Explanation of statistical techniques, no inclusion of the reasons to choose this technique | ANALYSIS:<br><br>• Errors presented of means and standard deviations/standard<br>• Description of number of cases with missing data and strategy to handle missing data<br>• Explanation of statistical techniques, reason of choosing and caveats |

Following Research Connections Quantitative Quality Assessment Tool (https://www.researchconnections.org/content/childcare/understand/research-quality.html, retrieved at 16 January 2019).

**Table 4.** Quality assessment criteria of included reviews.

| Low Quality | Medium Quality | High Quality |
|---|---|---|
| • No adequate formulation/description of research question<br>• No comprehensive literature search strategy<br>• No adequate description of selected study designs and justification of excluded articles<br>• No performing of study selection/data extraction in duplicate<br>• No appropriate methods for statistical combination of results<br>• Missing report of potential sources of conflict of interest<br>• No discussion about the heterogeneity of the studies in the results | • Not well-defined description of research question<br>• Literature search strategy is not comprehensive<br>• Exclusion/inclusion criteria are not specific<br>• Not well-defined study selection<br>• Methods used for statistical combination of results not adequately defined.<br>• Conflict of interest is not well assessed<br>• Insufficient discussion of any heterogeneity observed in results of review | • Adequate formulation and description of research question<br>• Specific and predefined exclusion/inclusion criteria<br>• Comprehensive/systematic literature search strategy (if systematic review)<br>• Independent review of titles, abstracts and full-text articles<br>• Using a standard method to appraise internal validity of included studies<br>• Assessment of publication bias and heterogeneity<br>• List and description of included studies<br>• Appropriate methods for statistical combination of results |

Following National Heart, Blood and Lung Institute Study Quality Assessment Tool. (https://www.nhlbi.nih.gov/health-topics/study-quality-assessment-tools, retrieved 16 January 2019).

### 2.6. Data Analysis

Meta-analysis was not conducted in this systematic review because the measurement units and effect sizes from different papers were not comparable. Some results from the papers are mentioned in this review but no graphs, forest plots or pictures are presented here. Instead, a narrative synthesis has been chosen.

## 3. Results

### 3.1. Selection of Articles and Studies

From four databases, 1246 results were obtained through the systematic literature research (see Section 2.1). After removing 352 duplicates, 894 article titles were screened for relevant content, yielding 68 articles deserving of closer reading were identified. After the full-text reading of these, 34 articles were excluded because their focus turned out to be tangential to our criteria. Thus, 34 articles have been included in our systematic review. The screening process can be seen in Figure 1.

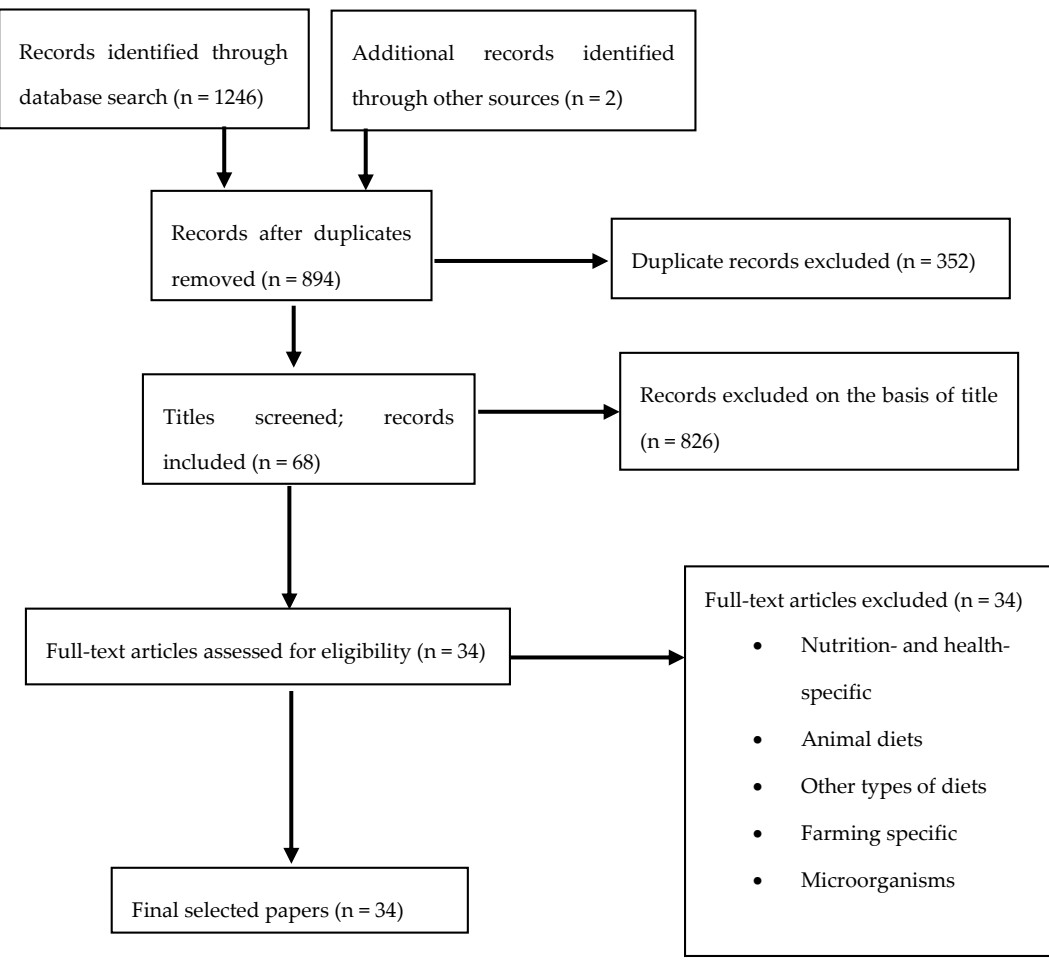

**Figure 1.** Preferred Reporting Items for Systematic Reviews and Meta-Analyses (PRISMA) diagram showing the screening process (http://www.prisma-statement.org).

### 3.2. Characteristics of Relevant Articles and Studies

Both the LOV and vegan diets will be referred to below as plant-based diets.

### 3.3. Environmental Impact of an Omnivorous Diet

The environmental impact of OMN production can be measured in several ways, of which we focus on three: GHGEs, land use and water footprint.

### 3.3.1. Greenhouse Gas Emissions

Regarding the carbon footprint, red meat production normally generates 23% of agriculture-related GHGEs [10]. Nitrous oxide ($NO_2$), methane ($CH_4$) and carbon dioxide ($CO_2$) are considered as GHGEs. GHGEs from agriculture induce changes in land use, especially deforestation, and are expected to increase [11]. Therein, $NO_2$ and $CH_4$ from livestock account for 80% of all agricultural GHGEs [2]. In the US, about 31% of $CH_4$ emissions are generated from enteric fermentation (primarily cows) and manure management. Emission per unit of livestock product varies with animal types. Apparently, GHGE emission is much greater for ruminant animals such as cattle, sheep and dairy, than for pigs or poultry [2]. It is estimated that about 44% of total global methane emissions are from livestock, and that the output is dominated by beef production. On average, 43 kg of GHGEs released during the production of each kg of beef. Of these 43 kg, approximately 22 kg are methane emissions. This result does not include the GHGEs from the beef carcass [12].

A great share of GHGEs are produced while food is in the supply chain, prior to final consumption. Meat production generates far more GHGEs than production for vegan and LOV diets [13]. Another study shows that, for each gram of beef protein consumed in the human diet, beef production requires 42 times more land use, 2 times more water use and 4 times more nitrogen, while it generates 3 times more GHGEs than the staple plant foods [12]. Finally, a study from India indicates that mutton and milk production contribute up to 23% and 35% of total local agricultural GHGEs, while all other food production combined contribute 16% of it [14].

The production of livestock contributes heavily to the GHGEs. These GHGEs are responsible for climate change and represent a real threat to our planet [2]. Another study estimates that meat and dairy production processes account for 80% of all GHGEs from the food sector and 24% of total GHGEs [2]. Meat and cheese production contribute around 40% to daily GHGEs [7].

It has been reported that ruminant animals emit the largest GHGs content per gram of protein and kcal. Methane comprises the largest part of GHGs. Based on self-reported dietary patterns in the UK, high meat consumers were responsible for 1.9 times and average meat consumers were responsible for 1.5 times more GHGEs than people on LOV diets [12], and 2.5 and 2 times more GHGEs, respectively, than vegan consumers [12]. One study concludes that, if the beef, dairy, pork, poultry and eggs consumed in an average European diet were reduced by 50% of and replaced with a 50% higher bread intake, the amount of GHGEs emitted could be reduced by over one-third [15].

Differences in the type of meat consumed can also be seen with multiple studies supporting an increase in GHGEs in diets with high amounts of ruminant meat consumption [16,17]. Walker et al. shows a comparison between environmental impact and the quality of the diet. It suggests that a reduced consumption of animal-based products, and an increased vegetable intake show lower GHGEs [18]. This statement is supported by multiple other studies with diets having an increased environmental impact when increasing the animal-based food intake [15,19]. Diets with low meat and low processed food consumption have lower GHGEs than their counterparts [20].

### 3.3.2. Land Use

Beef production requires a vastly larger amount of resources than the staple plant foods such as rice, beans, and potatoes. One study [2] calculates that each kg of beef requires 163 times more land use, 18 times more water use, 19 times more nitrogen and 11 times more $CO_2$ than 1 kg of rice or I kg of potatoes. A second study [7] points out that meat production accounts for 39% of land use related to human diet [7]. Moreover, compared to a LOV diet, the inputs needed to produce a non-vegetarian diet are: 2.9 times more water, 2.5 times more primary energy, 13 times more fertilizer and 1.4 times

more pesticides [7]. Furthermore, livestock farming uses 70% of agricultural land overall and a third of arable land. As such, it plays a major role in $CO_2$ release and biodiversity loss from deforestation [21].

Not all types of meat have the same protein conversion efficiency: chicken broilers have a conversion efficiency of 18%, while pork and beef have 9% and 6%, respectively [22]. This leads to a difference in land requirements between the animal-based protein sources, as more feed is required for the same amount of animal-based protein. Compared with soybean as a protein source, it becomes clear that land requirements are 6–17 times higher for animal-based protein [22].

### 3.3.3. Water Footprint

Livestock farming also generates water shortage. It largely uses finite irrigation water to supply the increasing demand for livestock products [21]. It is reported that animal production accounts for 12% of all groundwater and surface water used for irrigation. Therefore, the total water footprint equals 29% of the global agricultural production. One study determined that a diet containing a lower volume of livestock products would result in reduced global water consumption. Water input depends on the season and annual fluctuations in rainfall. More water is used for meat production than for plant protein production. One study finds that the difference between water inputs for animal protein vs. plant protein is normally around a factor of 26; even when intensive irrigation is needed for plant-based protein, animal protein production requires 4.4 times as much water [22]. A second study supports this finding, stating that production for LOV diets has increased the water-scarcity footprint by 26% [15]. Nevertheless, it is difficult to make general scientific claims, since studies regarding metrics of water use are based on very limited evidence [23]. Producing each kg of consumable beef requires about 13 kg of grain and 30 kg of hay, which in turn require 105,400 L of water [24]. Furthermore, 500–2000 L of water are required to produce one kilogram of crop [24]. In terms of fossil energy used in the whole process, the input needed to produce 1 kcal of plant protein is 2.2 kcal. For instance, one kg of protein obtained from a plant-based source requires approximately 100 times less water than one kg of protein from an animal origin [12].

### 3.4. *Environmental Impact of a Vegetarian Diet*

González-García et al. have studied variations in dietary patterns, including within vegetarian diets, around the world. They note that some LOV diets include fish (pesco-vegetarian diet also found in Denmark) and even meat on rare occasions (e.g., the "semi-vegetarian diet," which is found in India), and they find that the major differences between these diets are associated with calorie intake, specific food choice and national dietary guidelines [8]. Our review is limited to studies of LOV, which does not include any type of meat, but does include animal products such as milk, cheese and eggs.

### 3.4.1. GHGE Impacts for Production of LOV Diets

It is widely believed today that plant-based diets (such as vegetarian and vegan) have a positive impact on the environment and health, and they are indeed shown to have many benefits: safety for human consumption, waste management, storage options and lower GHGEs than the animal-based diets [8]. Although proteins from plant-based sources are considered to be of lower quality than proteins from animal origin, a well-planned plant-based diet can be both nutritionally sufficient and environmentally sustainable [25]. Preferred sources of plant-based proteins are quinoa, amaranth, wheat, pulses and soy-based products such as tofu and tempeh [8]. Milk, dairy products and eggs are another important source of proteins in LOV diets. A study in Sweden compared beef with soybeans and reports that per gram of protein, beef requires 18 times more energy and produces 71 times more $CO_2$ than soybeans [12]. A study of different dietary patterns in the UK determined that OMN had causes 4 times higher GHGEs per kcal than LOV. A vegan diet was not studied separately, but the authors conclude that dairy accounts for about 40% of GHGEs associated with production for LOV diets [12].

Vegetarians often use meat substitutes, but the different substitutes carry very different environmental implications. Depending on what substitute is used, the associated reduction in meat consumption can result in no positive effect on the environment at all or even trigger a negative trend [7]. One study indicates that, despite the likelihood that more consumers of vegetarian diets will lead to reduced GHGEs, this outcome is not guaranteed. For instance, substituting cheese for chicken in a diet could result in higher aggregate GHGE production if the energy and nutrient content is not considered and production of the vegetarian substitute is associated with higher GHGEs [26]. Other important factors are transportation (especially long-distance), deep-freezing, and some specific horticultural practices, all of which can lead to higher environmental damage than locally produced organic meat [22].

One of the most important issues in environmental studies are GHGEs, most notably carbon dioxide. Estimates of $CO_2$ emissions vary widely from study to study. The carbon footprint in Italy was found to be $2.60 \pm 0.62$ kg $CO_{2eq}$ per person, per day [27]. Another study presented carbon dioxide in terms of calories and calculated that the total amount produced by vegetarians and semi-vegetarians for average calorie intake of 2000 kcal was 3.81 kg $CO_{2eq}$ [7]. Nitrogen gasses are also very important factor. Nitrogen is usually added through fertilizers to improve crop yields. Its negative impact results in acid rain, biodiversity loss, stratospheric ozone depletion, climate change, eutrophication and smog [25]. Reported nitrogen footprint for the vegetarian diet is $18.3 \pm 2.4$ kg per capita per year [25].

### 3.4.2. Land Use for Production of LOV Diets

The land used to produce protein from plants is much lower than the production of proteins from animals. For example, for soybean production, the requirements are a factor of 6–17 larger for production of meat proteins (depending on the type of meat) [22]. In previous years, about 0.5 hectare (ha) of cropland was used in omnivorous diet and less than 0.4 ha for vegetarian-based diet [24]. It should be taken into account that, in order to produce meat, animals need to be fed plants. Another important thing is how much of animal-based products are incorporated in the vegetarian diet. If the diet is high-fat, more land is required [27]. A study from the USA by Blackstone et al. found that the vegetarian diet has the lowest GHGEs and is favorable when it comes to sustainability [28].

Overall, the consumption of animal products which presents a secondary production is much less efficient than eating plants which directly convert solar energy to food energy (primary production) [12].

### 3.4.3. Water Impact of LOV Diets

The water-related effects of agricultural production for LOV diets essentially parallel the discussion in Section 3.3.3 above, and need not be repeated here.

### 3.5. Environmental Impact of a Vegan Diet

Multiple studies have shown that diets rich in vegetables have a better influence on the environment than those rich in meat. It is therefore proposed, both as a conclusion of this systematic review and of several studies reviewed herein, that the vegan diet is the most sustainable diet in terms of environmental footprint e.g., [8].

One analysis shows that high meat eaters in the UK had 1.9 times and medium meat eaters had about 1.5 times more of GHGEs than an LOV, and that the food consumed by high meat eaters is associated with 2.5 times more GHGEs that that consumed by a vegan, and even average meat eaters are responsible for twice as many GHGEs [12]. A second study reports similar conclusions, based on the number and composition of 2000 kcal consumed in various diets: 7.19 kg for high meat eaters (100 g of meat per day or more), 5.63 kg for medium meat-eaters (50–99 g of meat per day), 4.67 kg for low meat-eaters (less than 50 g per day), 3.81 kg for vegetarians, and 2.89 kg for vegans [7,29].

Most studies demonstrate that, in general, vegan diets are the most environmentally sensitive. However, this some authors would disagree and would suggest that 100% plant-based food consumers may need larger volumes of food than vegetarians to achieve the same energy intake [27]. The main

reason, however, is that many vegans replace animal-based products with processed plant-based meat and dairy substitutes (e.g., seitan burger and soy yoghurt) instead of consuming the unprocessed, plant-based nutritious foods that are relatively favored in many LOV diets. For example, one study finds that vegetarians in the USA substitute meat mostly with dairy products and, to a lesser extent, with fruits, vegetables and oil [12], that is, with the foods that, aside from meat, have the most deleterious environmental impacts. These choices are described as the main reason why GHGEs associated with plant-based diets are not as low as they should be, and also highlights the importance of reducing dairy consumption in all diets. When dairy is reduced or eliminated, as it is LOV and vegan diets, these two diets produce 33% and 53% lower emissions for the same number of calories (2000 kcal) as the average US diet [12]. The production of vegan cheese-like spread (lupine-based cheese) requires one-fifth of the land required for cheese from cow's milk: 0.02 ha of land per 100 kg, compared with 0.1 ha of land per 100 kg of cow-milk-based cheese [22]. Consuming legumes for protein instead of meat has a beneficial environmental impact, and it is also a lot cheaper [25].

A life cycle assessment analysis suggests that, if beans were substituted for beef, then 692,918 km$^2$ of US cropland could be freed up for other uses and GHGEs from this land would decrease by 74% [30]. Perignon et al. states that if a replacement of all meat and dairy products by plant-based food would take place, land use could be reduced by 50% [7].

A large part of plant-based diets consists of fruits and vegetables. The origin and mode of transport of fruits and vegetables has a big impact on their contribution to GHGEs, which can vary a lot. Whether they are produced in heated greenhouses or not has a huge effect on their GHGEs [31]. Locally grown and sold fruits and vegetables have been assumed to be more environmentally friendly. However, it has been shown that this might not be the case. A study shows that when customers in the UK choose to drive more than 7.4 km to buy locally grown fruits and vegetables, the GHGEs would be higher than if a large-scale delivery system transported the food closer to the customer [12].

Nonetheless, one study estimates that a complete switch to a vegan diet could result in reductions of 17% for $CO_2$, 21% for $NO_2$ and 24% for $CH_4$ [2]. Among the three diets, the vegan diet makes the lightest demands on the global water supply, requiring 14.4% less freshwater and 20.8% less ground water than the omnivorous diet [12].

## 4. Discussion

The general outcome that can be concluded from this review is that the more plant-based a diet is, the more sustainable. However, in some cases, the vegan diet may not have a lower environmental footprint than LOV. The reason for this is that vegans tend to replace animal-based products in their diet by industrially, highly processed plant-based meats and dairy substitutes [27].

This literature review looked into three main indicators of the environmental impact caused by three types of human dietary patterns: GHGEs, the use of land and the water footprint. The studies reviewed here were selected by use of a specific inclusion and exclusion process, which is explained in the Methods section. Studies that focus directly on environmental impacts of human diets were considered most relevant for this review.

This research involves some limitations. First of all, the method used allowed for only one search, with the same keywords in each database. As a result, certain articles that are relevant to the subject of the paper may have been overlooked. The choice of search words could also have placed undesired limits on the results generated. Keywords such as 'environmental assessment' and 'life cycle assessment' for example, have not been included in the literature search. For this reason, some high-quality studies are inevitably missing from this paper. Secondly, all papers that survived through the title screening were carefully examined by all authors before the final exclusion took place, to ensure no loss of data (see Section 3.1). Still, the possibility remains that the collective view of the authors might be different from that of a reader. Besides these technical limitations, this review is limited to assessing the relationship between three diets and only three among the many environmental factors that are affected by dietary choices. We justify this choice on the basis of our

claim that GHGEs, land stress and water supply are the biggest causes of environmental damage, broadly understood. Obviously, more environmental factors could have been taken into account, but doing so would have risked drawing attention away from our main purpose. A final notable limitation is the geographic scope of the papers included. Most of the papers are from high-income countries, and several others including global data. Thus, we are unable to assess the environmental consequences of food production in low-income countries. We are sensitive that, due to variations in input intensity and production efficiency, for example, these consequences may differ markedly from those in high-income countries [31].

A quality assessment table for the included studies can be found in the method section. This table shows what terms each report needs to meet in order to be classified as low, medium or high quality. The assessed quality of each paper is listed in Tables 1 and 2. The quality assessment further allows to state that the narrative synthesis performed in this paper is sustained by a majority of papers and reviews with high or medium quality.

Although it was not the main focus of this research, food waste is another important contributor to climate change, as the production of every kg of unconsumed food entails has the same environmental impact as a kg of consumed food. As with consumption, plant-based diets are also more climate friendly when they are wasted. One study, conducted in the US, finds that fruits and vegetables which comprise 33% of food waste, account for only 8% of carbon dioxide emissions. Animal-based foods, by contrast, account for 33% of food waste by mass and 74% of carbon dioxide emissions. Ruminant meat accounts for 3% of waste by mass and 31% of emissions from waste. Thus, in order to prevent increased agricultural expansion, must be reduced [12]. Besides food waste, it is also important to note that food miles contribute heavily to the GHGEs associated with a specific diet. A plant-based diet that requires products from all over the world will have a footprint equivalent to a moderate meat eater. Reduction of GHGEs per household on a fully local diet is equivalent to 1600 km/year driven and therefore an important factor in mitigating GHG from diets [4]. A British study on the environmental impact of food transport, in this case on the effect of potatoes, rice and pasta on GHGEs and water use, shows that the environmental impact of transporting food products from the same group can vary significantly [32].

According to the results presented here, it is very clear that the vegan diet has the least environmental impact in comparison with LOV and OMN diets. Vegans do not consume any animal products and thus are able to avoid all the negative environmental impacts that these animal-based products bring. It is very important to note that vegan products that are highly processed, high in fat, or have to be transported long distances may have considerably larger environmental footprints. Possible negative impacts of LOV and vegan diets on human health should also be assessed. Insufficiencies of vitamin B12, calcium, iron and other nutrients could appear if diets are not well balanced [33]. Still, a 2014 study demonstrates that synergies can be developed between a healthy diet and an environmentally sustainable food pattern [34].

Healthy OMN diets come in many forms; some currently popular variations include the Atlantic diet, the Mediterranean diet and the Nordic diet. All of these diets differ—both from each other and from other OMN diets that include a higher proportion of beef products—in their animal-based protein sources and the amount of animal-based protein consumed. This complicates efforts to estimate precisely the environmental impact of the OMN diet conceptualized as a discrete entity. Results show that the OMN diet has the highest environmental footprint mainly due to the consumption of animal flesh, dairy products and eggs. In very rare cases it is possible for an OMN consumer to be more sustainable than a consumer following a LOV, e.g., when a LOV diet including large amounts of highly processed foods, imported from afar is compared with an OMN diet that follows national dietary recommendations that urge moderate meat, dairy and eggs consumption mostly from local sources. If consumers would follow current dietary recommendations e.g., for Mediterranean and Nordic diets, their corresponding weekly GHGEs resulting of the consumption of food rich in protein (meat, legumes, milk, etc.) would have a comparable and similar impact on the environment as consumption

of vegetables and fruits [1]. Additionally, adherence to recommendations will have similar impact to plant-based diets (vegan or vegetarian) as the common denominator for all recommendations is to increase consumption of foods of plant origin while reducing those of animal origin [35] or highly processed [36].

Even though results of this systematic review point to a 100% plant-based diet such as the vegan diet as the best solution for the future, such changes are hard to achieve at population level if the recommendations clash with cultural expectations and norms. For example, linear programing can be used to identify a dietary pattern that is healthy and similar in its environmental impact to the vegan diet, but based on the current foods consumed by the population. This method could identify possible diets with the same impact on the environment as vegan diet but with lesser change from the original diet in other cultural contexts [25,37].

## 5. Conclusions

The present review based on 16 papers of high quality, 9 of medium quality and 9 of low quality, shows a consistent and clear difference between the environmental impacts of different diets. The GHGEs differ considerably per diet, with a vegan diet having the lowest $CO_{2eq}$ production per 2000 kcal consumed.

The environmental impact on land and water also differs among the three diets. Water use is higher in LOV and OMN diets, due to the use of animal-based proteins. In short, the more animal protein consumed in a diet, the higher the water use will be. A diet pattern based only of foods of plant origin offers the greatest potential for reduced global water consumption. Furthermore, livestock farming uses 70% of agricultural land overall and a third of arable land. On this account, a vegan diet has the lowest land use and water use of the three different diets.

In conclusion, a 100% plant-based diet (e.g., vegan) has the least environmental impact. Therefore, this review further supports the wealth of existing evidence supporting a transition to a more sustainable food system and food consumption. Still, it is important to note that, in order for a 100% plant-based diet to be sustainable, local products that minimize the environmental impact of transport should be preferred. Further research should focus on the GHGEs from different types of plant-based foods, and modified omnivorous diets with the same environmental impact as the impact from vegan diet.

**Author Contributions:** Conceptualization, all authors; methodology, B.C.C., J.R.v.d.V., K.G., H.G.E., I.K.; software, B.C.C.; formal analysis and data extraction, B.C.C., J.R.v.d.V., K.G., H.G.E., I.K.; writing—original draft preparation, B.C.C., J.R.v.d.V., K.G., H.G.E., I.K.; writing—review and editing, B.C.C., F.J.A.P.-C.; visualization, B.C.C., F.J.A.P.-C.; supervision, F.J.A.P.-C.

**Funding:** This research received no external funding.

**Conflicts of Interest:** The authors declare no conflicts of interest.

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
