# Peer review of "Which Diet Has the Least Environmental Impact on Our Planet? A Systematic Review of Vegan, Vegetarian and Omnivorous Diets"

_sustainability, doi:10.3390/su11154110_

Reviewer 1 Report

The authors present a systematic review assessing the environmental impact of vegan, vegetarian, and omnivorous diets. This is an interesting question, and I applaud the researchers for investigating it. However, there are several notable issues that raise concern. The authors did not adequately review the literature for all relevant studies, because several important ones are missing. And in general, the paper reads more like an argument against animal-based foods rather than an unbiased scientific pursuit. There are also a number of important indicators of sustainability, and GHG emissions, land footprint, and water use are only several of them, so it is not fair to make a general conclusion about sustainability given this limited suite of indicators. Additional comments can be found below.

Introduction

This section does not provide a sufficiently wide review of the literature, and relies on only several (mostly outdated) sources.

Please have a native English speaker review this manuscript.

Methods

If the objectives of the paper were to assess GHGs, water footprint, and land footprint, then “water” and “land” should be included in the search terms.

Results

The way the sub-headings are written, it implies that environmental impact had an effect on the diet, when the opposite is true. Please make this change.

This section reads as if it were an argument against consuming meat, rather than an unbiased and scientific analysis of the research question at hand.

Discussion

This section does not follow a standard structure used in peer-reviewed scientific journal articles. For example, the limitations should not be listed in the first paragraph, and the main conclusions should not be listed in the third paragraph.

Author Response

Introduction

This section does not provide a sufficiently wide review of the literature, and relies on only several (mostly outdated) sources.

The introduction has now been amended with papers from 2019 contextualising the reason why this review is pertinent in the current context.

Methods

If the objectives of the paper were to assess GHGs, water footprint, and land footprint, then “water” and “land” should be included in the search terms.

We thank the reviewer for this suggestion. The authors have performed a search adding the terms water and land in the search terms. However, no additional documents meeting the inclusion criteria were found.

Results

The way the sub-headings are written, it implies that environmental impact had an effect on the diet, when the opposite is true. Please make this change. 

Thanks for the recommendation. We have made the change as requested.

This section reads as if it were an argument against consuming meat, rather than an unbiased and scientific analysis of the research question at hand.

We believe that the way this section is presented reflects the data gathered in the review. We have been careful to extract data in a systematic way, we have used triangulation (three or more authors agreement) in the analysis and interpretation of the data in order to avoid any kind of reflexivity bias.

Discussion

This section does not follow a standard structure used in peer-reviewed scientific journal articles. For example, the limitations should not be listed in the first paragraph, and the main conclusions should not be listed in the third paragraph.

We thank the reviewer for this suggestion, and it has been implemented now.

Reviewer 2 Report

Thank you for providing an interesting study on assessments of diets from a European and USA perspective. I find the results rather well presented and the methods described, but I have found some serious shortcomings with the study.

The keywords used, and their motivation, are rather vague. As the aim of the study is to review envirronmental impacts, it would be more appropriate to include keywords such as environmental assessment, life cycle assessment, etc. Limiting only to GHG or footprint to review environmental impacts is a major limitation of the study.  This should at least be identified as there are many studies from Europe (e.g. only in Sweden) from authors such as Hallström et al., Röös et al., and a recent study in this journal from Martin and Brandao reviewing the environmetal impacts of diets.

How were the "low, medium and high-quality" assessed/categorized?

The text is quite repetitive, reviewing different diets and the ranges of figures. It would be important to show (e.g. with bar charts) the ranges in the different studies outlined (even if done for different countries). How much better were vegan/veg. diets compared to normal. 

Reviewer 3 Report

The topic is very actual and, even if environmental issues are often considered in the plastic waste, industrial emission and fossil fuel usage, I think that nutrition is a promising intervention with much more feasibility than alternative materials and fuel utilization.

The data are noteworthy and interpretation is prudent and coherent with evidence in the literature. However, the manuscript needs some improvement that I suggest below:

-Diet classification must be more rigorous clearly shown. For example, at the beginning of the results, a diversification among vegan, lacto-ovo-vegetarian is needed, adopting “vegetarian” term as the equivalent of plant-based. You can take inspiration from your first reference in order to reduce the misunderstanding of the generical “vegetarian” term.

-The results sections for 3.3 to 3.5 are confusing and often data from the literature regarding a single diet are not displayed in relation to other diets (for example, at lines 282-208. It could be better to organize results not per diet but per emission or environmental topic (lands, water, GHGE, etc.), by comparing the three main diets.

-In a manuscript regarding environmental impact and food, the 2013 FAO document “Tackling climate change through livestock” cannot miss. If the data from this document are not usable in data analysis of your manuscript, then still deserves to be reported in the introduction and/or discussion sections.

Other minor issues are listed below:

-Lines 20-21 and 49-51 seems to be unnecessary. The title of the manuscript is clear and well-focused to the topic.

-At line 71-72, if a reference were excluded from the main analysis, it is unnecessary to report it as an example.

-“Ha” abbreviation needs to be explained at the first use, even if it is obvious, as all abbreviations in the paper.

-At line 355 there is no need to specify “meatless”, in particular, if it was explained before.

-Even if it is somewhat off-topic in respect to this paper, a little suggestion for the use of insect for feed and food could complete the discussion about alternative solutions for the ecological resolution and for a future discussion.

Round  2

Reviewer 2 Report

Thank you for taking into account the comments that I provided. I could not find the report for the other reviewers, but I find your rebuttal adequate.

I still wonder why studies such as Martin and Brando, 2017 (Evaluating the Environmental Consequences of Swedish Food Consumption and Dietary Choices) is not included given the keywords used. Especially important to reference articles from this journal if possible.

After careful consideration, I would have liked to see the above article included, at least in the discussion. Otherwise, I have only some minor comments for the paper.

1) Why are page numbers used after the references? This should be used e.g. when direct quotes are used to refer to a page, etc. 

2) Line 78-79....why is this added as a separate study, wrong reference style?)

3) Figure 1 arrows are a bit overlapping with the boxes

4) When reviewing the references, there is something wrong. The "Author 1, Author 2" is showing up. This needs to be resolved. 

5) in your keyword search, remove the second instance of "environment" you can see it here:

AND ( footprint OR sustain* OR impact ) AND ( "greenhouse gas*" OR vegetarian* OR vegan

62 OR omnivorous ) ( environment* )

6) Line 394, lower case for the word complicate

7) In the beginning of the conclusion section, the terms for the diets should be revised to state "a...xxx...diet" instead of "the" since it is hard to know what "the" is.

Author Response

Thank you for taking into account the comments that I provided. I could not find the report for the other reviewers, but I find your rebuttal adequate. 

-Thank you for your overall assessment of the rebuttal.

I still wonder why studies such as Martin and Brando, 2017 (Evaluating the Environmental Consequences of Swedish Food Consumption and Dietary Choices) is not included given the keywords used. Especially important to reference articles from this journal if possible. 

After careful consideration, I would have liked to see the above article included, at least in the discussion. Otherwise, I have only some minor comments for the paper. 

-Following your suggestion, the article under reference 43 has been added as part of the discussion.

1) Why are page numbers used after the references? This should be used e.g. when direct quotes are used to refer to a page, etc. 

-The document has been amended accordingly. It was our misinterpretation of the guidelines. Thanks for this.

2) Line 78-79....why is this added as a separate study, wrong reference style?)

-Done

3) Figure 1 arrows are a bit overlapping with the boxes

-Done

4) When reviewing the references, there is something wrong. The "Author 1, Author 2" is showing up. This needs to be resolved. 

-Done

5) in your keyword search, remove the second instance of "environment" you can see it here: AND ( footprint OR sustain* OR impact ) AND ( "greenhouse gas*" OR vegetarian* OR vegan 62 OR omnivorous ) ( environment* )

-Done

6) Line 394, lower case for the word complicate

-Done

7) In the beginning of the conclusion section, the terms for the diets should be revised to state "a...xxx...diet" instead of "the" since it is hard to know what "the" is. 

-Done

Reviewer 3 Report

I think the manuscript was implemented compared to the previous version and the authors accepted my suggestions for the revisions. However, some aspects remain to be improved, mostly graphic misprints. I also recommend a language revision to improve readability.

Following are my suggestions:

- For consistency with the definition already accepted and inserted at lines 49-54, the title of the manuscript also would need correction in this regard. More generally, with the exception of the search criteria adopted for literature revision, would be preferable to use the lacto-ovo-vegetarian (LOV) and plant-based nomenclatures, avoiding the term "vegetarian" also in the text (for example on lines 159, 212, 229, 230, 238, etc.). Choose LOV or plant based on the need

- I don't think it's necessary to insert the page reference after the bibliographic reference in the text. If it is not present in the journal guidelines for authors, I suggest removing them.

- In the final list of references, there are numerous graphic typos indicated as author 1.2, etc. Perhaps they are errors of the application used for managing references. Please correct these typos.

- At the end of line 31, the dot is missing.

- At line 49 "soil, land effects" give the impression of being two parameters. Better to prefer a unique nomenclature of the environmental indicator such as "land use". More generally, it is better to always use the same term (for example on lines 49, 186, 270

- The same applies to the term GHGEs (for example to lines 169, 171, 378)

- At line 85 the phrase "this was done" does not seem necessary.

- Figure 1, the first two arrows of the diagram must be arranged due to an overlap error.

- It is better to standardize the “water footprint” indicator therm at the line 145 lines (water use) and line 201 (water shortage).

- The qualitative criteria used together with references should also be added in the methods paragraph and not only in tables 3 and 4.

- The period at line 164 is not very clear and percentages do not seem consistent.

- At line 171 it could miss a point before the term "Based".

- At 201 the term pollution is off-topic and in any case, not correctly treated.

- Lines 220-227 are repeated and must be removed.

- At line 262 replace "Italian" with "Italy". However, the period 262-263 is not clear why it is presented.

- The period at 306-307 should be reformulated because it is unclear.

- At line 329 there is an error in NO2 (and not N2O).

- The period at lines 343-344 must be reformulated because it is unclear.

Author Response

I think the manuscript was implemented compared to the previous version and the authors accepted my suggestions for the revisions. However, some aspects remain to be improved, mostly graphic misprints. I also recommend a language revision to improve readability.

-Thanks for your evaluation of the improvements. The manuscript has been sent to a professional proofreader and native English speaker. 

Following are my suggestions:

For consistency with the definition already accepted and inserted at lines 49-54, the title of the manuscript also would need correction in this regard. More generally, with the exception of the search criteria adopted for literature revision, would be preferable to use the lacto-ovo-vegetarian (LOV) and plant-based nomenclatures, avoiding the term "vegetarian" also in the text (for example on lines 159, 212, 229, 230, 238, etc.). Choose LOV or plant based on the need

-Done

I don't think it's necessary to insert the page reference after the bibliographic reference in the text. If it is not present in the journal guidelines for authors, I suggest removing them.

The MDPI reference guide suggests us to insert page number after the reference number. We have however removed the page references in the text.

In the final list of references, there are numerous graphic typos indicated as author 1.2, etc. Perhaps they are errors of the application used for managing references. Please correct these typos.

-Done

At the end of line 31, the dot is missing.

-Done

At line 49 "soil, land effects" give the impression of being two parameters. Better to prefer a unique nomenclature of the environmental indicator such as "land use". More generally, it is better to always use the same term (for example on lines 49, 186, 270

-Done

The same applies to the term GHGEs (for example to lines 169, 171, 378)

-Done

At line 85 the phrase "this was done" does not seem necessary.

-Done

Figure 1, the first two arrows of the diagram must be arranged due to an overlap error.

-Done

It is better to standardize the “water footprint” indicator therm at the line 145 lines (water use) and line 201 (water shortage).

-Done

The qualitative criteria used together with references should also be added in the methods paragraph and not only in tables 3 and 4.

-We have added the sentences in the text.

The period at line 164 is not very clear and percentages do not seem consistent.

-Done

At line 171 it could miss a point before the term "Based".

-Done

At 201 the term pollution is off-topic and in any case, not correctly treated.

-Done

Lines 220-227 are repeated and must be removed.

-Done

At line 262 replace "Italian" with "Italy". However, the period 262-263 is not clear why it is presented.

The period at 306-307 should be reformulated because it is unclear.

At line 329 there is an error in NO2 (and not N2O).

-Done

The period at lines 343-344 must be reformulated because it is unclear.

-Done

This manuscript is a resubmission of an earlier submission. The following is a list of the peer review reports and author responses from that submission.